# Toxicogenomic Effects of Dissolved Saxitoxin on the Early Life Stages of the Longfin Yellowtail (*Seriola rivoliana*)

**DOI:** 10.3390/md21110597

**Published:** 2023-11-18

**Authors:** Colleen Guinle, Erick Julián Núñez-Vázquez, Leyberth José Fernández-Herrera, Daniela Alejandra Corona-Rojas, Dariel Tovar-Ramírez

**Affiliations:** 1Centro de Investigaciones Biológicas del Noroeste, Laboratorio de Fisiología Comparada y Genómica Funcional, Av. Instituto Politécnico Nacional 195 Playa Palo de Santa Rita, La Paz 23096, Mexico; colleenguinle@gmail.com (C.G.); daniela.corona.rojas.2020@gmail.com (D.A.C.-R.); 2Centro de Investigaciones Biológicas del Noroeste, Laboratorio de Toxinas Marinas y Aminoácidos, Av. Instituto Politécnico Nacional 195 Playa Palo de Santa Rita, La Paz 23096, Mexico; lfernandez@pg.cibnor.mx

**Keywords:** saxitoxin, paralytic shellfish toxins, toxicogenomic, gene expression, *Seriola rivoliana*, harmful algal blooms, embryonic development, hatching percentage

## Abstract

Harmful algal blooms (HABs) can produce a variety of noxious effects and, in some cases, the massive mortality of wild and farmed marine organisms. Some HAB species produce toxins that are released into seawater or transferred via food webs (particulate toxin fraction). The objective of the present study was to identify the toxicological effects of subacute exposure to saxitoxin (STX) during embryonic and early larval stages in *Seriola rivoliana*. Eggs were exposed to dissolved 19 STX (100 μg L^−1^). The toxic effects of STX were evaluated via the hatching percentage, the activity of three enzymes (protein and alkaline phosphatases and peroxidase), and the expression of four genes (*HSF2*, *Nav1.4b*, *PPRC1*, and *DUSP8*). A low hatching percentage (less than 5%) was observed in 44 hpf (hours post fertilization) embryos exposed to STX compared to 71% in the unexposed control. At this STX concentration, no oxidative stress in the embryos was evident. However, STX induced the expression of the NaV1.4 channel α-subunit (*NaV1.4b*), which is the primary target of this toxin. Our results revealed the overexpression of all four candidate genes in STX-intoxicated lecithotrophic larvae, reflecting the activation of diverse cellular processes involved in stress responses (*HSF2*), lipid metabolism (*PPRC1*), and MAP kinase signaling pathways associated with cell proliferation and differentiation (*DUSP8*). The effects of STX were more pronounced in young larvae than in embryos, indicating a stage-specific sensitivity to the toxin.

## 1. Introduction

Harmful algal blooms (HABs) have become a major environmental problem in recent decades due to their impacts on fisheries, aquaculture, wildlife, socioeconomic and recreational activities, and human health [1]. Some HAB species produce marine toxins, which are released into seawater or transferred through food webs (particulate toxin fraction). The nature of these effects depends on the marine toxins produced by harmful microalgae (e.g., dinoflagellates, diatoms, and raphidophytes), cyanobacteria, and bacteria. These toxins are metabolic compounds that can accumulate in the tissues of marine organisms, affecting both their physiology and behavior [2,3]. For example, HABs have been responsible for mass mortality events in the Gulf of California of wild (e.g., mammals, birds, fish, and invertebrates) and farmed (e.g., fish and crustaceans) marine species due to the large quantities of toxins in contaminated prey [4,5]. In addition, many human poisonings have been linked to the consumption of seafood products (filter-feeding bivalves and fish) contaminated by toxins and cases of respiratory tract irritation due to aerosolized toxins [3,6,7].

Saxitoxin (STX) and its derivatives, which are commonly referred to as paralytic shellfish toxins (PSTs), are among the most studied marine toxins responsible for poisonings in both humans and marine animals [8,9]. These toxins can induce neuromuscular symptoms such as muscular paralysis, myalgia, and respiratory difficulty. In cases of severe intoxication, muscular paralysis or dyspnea may evolve into respiratory arrest and death [9,10]. Paralytic shellfish toxins act in the nervous systems of mammals by blocking voltage-gated sodium channels (Na_V_) [11,12] and can also bind to voltage-gated calcium (Ca_V_) and potassium (K_V_) channels [13,14]. Although PSTs have been linked to the mass mortality and intoxication of diverse marine organisms, including crustaceans, fish, sea birds, sea turtles, and marine mammals, knowledge of the corresponding action mechanisms and intoxication routes remains scarce [15].

The toxins released during HAB events accumulate in food webs. Thus, marine fish are the primary group affected by these toxins. In fish, as in other vertebrates, embryonic development is controlled by different signaling pathways in which specific molecules play important roles in various processes, such as dorsal–ventral and anterior–posterior positioning [14]. Marine toxins are capable of deregulating gene expression and can disrupt embryonic development by affecting cellular metabolism [16,17] and inducing developmental delays and macroscopic organ alterations or abnormalities [18,19,20,21]. Fish embryos and larvae are susceptible to marine toxins because they have not yet developed efficient detoxification systems and exhibit high metabolic growth rates [22]. Thus, these toxins, which have lethal and sublethal effects during early life stages [4,19,20], threaten fish recruitment, fishery yields, and aquaculture production [5,23,24]. Therefore, strategies must be implemented to detect and understand the mechanisms by which marine toxins affect fish and ultimately threaten human health.

The longfin yellowtail (*Seriola rivoliana*) is a carnivorous, benthopelagic marine fish of the family Carangidae that is mainly found in subtropical reefs from the United States of America (USA) to Peru in the Pacific [25] and from the United States to Argentina in the Atlantic [26]. *Seriola rivoliana* is a top carnivore in reef-associated food webs and the main vector of Ciguatera fish poisoning in the Atlantic (Macaronesian) Islands (Canarias and Madeira archipelagos) and Azores [27,28,29]. As with other *Seriola* species, *S. rivoliana* exhibits excellent potential for aquaculture production due to its worldwide distribution, fast growth, and ability to adapt well to captivity [30]. Indeed, *S. rivoliana* has become one of the most important marine fish species selected to diversify aquaculture production in Japan, Australia, the United States, Spain, and Mexico [30,31]. However, information on the early life stages of *S. rivoliana* is needed to elucidate the causes of early mortality and develop strategies to increase egg viability and larval survival [30,32,33].

The objective of the present study was to identify the effects of saxitoxin, one of the most toxic PSTs and one of the main PSTs present in Mexico. The use of marine fish embryos and larvae as sentinels to detect the modes of action of marine toxins has been proposed in many studies mostly conducted with Zebrafish (*Danio rerio*) [19,20]. Previous studies have reported the negative effects of toxins on fish larvae [34,35]. In *S. rivoliana*, exposure to diarrhetic shellfish toxins (DSTs; okadaic acid [OA] and dinophysistoxin-1 [DTX-1]), PSTs, and saxitoxin analogs (gonyautoxin 2–3 [GTX 2–3], decarbamoyl gonyautoxin 2–3 [dc-GTX2–3], and C1–C2) has been recently shown to deregulate embryogenesis-related pathways, inhibit phosphatase activity, and induce metabolic responses in larvae by increasing gene expression and lipid metabolism enzyme activity [36,37].

The impacts of STX exposure were also evaluated in larvae from hatching until 62 h post fertilization (hpf; i.e., before mouth opening at 96 hpf) [38]. The embryotoxic effect (hatching percentage) was quantified, and the regulation of genes and enzymes involved in (i) stress responses (heat-shock proteins), (ii) ion transport (voltage-gated sodium channels), (iii) lipid metabolism (peroxisome proliferator-activated receptors), (iv) digestion-related pathways (alkaline phosphatases), (v) oxidative stress (peroxidase), and (vi) key enzyme regulation (protein phosphatases) was also assessed. The results of the present study provide new information on the modes of action of PSTs during embryonic and larval development.

## 2. Results

### 2.1. Hatching Percentages

In the present study, *S. rivoliana* embryos and recently hatched larvae were used as models to qualitatively and quantitatively assess the impacts of PSTs during the development of early life stages. We hypothesized that PSTs would induce adverse effects on the development of *S. rivoliana* embryos, which could be evaluated with biochemical and molecular methods. To test this hypothesis, fish embryos were exposed to STX throughout embryonic development (8 to 44 hpf). The cellular and molecular levels in *S. rivoliana* were used to develop models with improved sensitivity to detect, quantify, and classify the modes of action of these molecules. The hatching percentage was assessed at 44 hpf in the control and STX treatment. The observed spawning volume (Appendix A), buoyancy, and shape indicated good egg quality. This preliminary observation was confirmed by the relatively high hatching percentage of eggs from the control group (71.16 ± 1.97%). Conversely, high mortality of STX-exposed eggs was observed in the different replicates, and the hatching percentage was 14 times lower (4.92 ± 0.80%) compared to that of the control group, with the difference being highly significant (Student’s *t*-test, *p* = 0.00018; Figure 1).

### 2.2. Gene Expression Analyses

The expression of heat shock factor 2 (HSF2), sodium channel protein type 4 subunit alpha B (*NaV1.4b*), peroxisome proliferator-activated receptor gamma coactivator-related protein 1 (*PPRC1*), and dual specificity phosphatase 8 (DUSP8) genes was detected in the control from 8 to 62 hpf. The highest expression levels were observed at 50 hpf. For the STX treatment, the expression of *HSF2*, *PPRC1*, and DUSP8 genes was detected from 26 to 62 hpf. In comparison, *NaV1.4b* gene expression was only detected between 50 and 62 hpf, which corresponded to the lecithotrophic larval stage (i.e., depletion of the remaining yolk reserves before mouth opening; Figure 2).

A significant increase in *HSF2* and DUSP8 gene expression was observed between 50 and 62 hpf in control embryos (*p* < 0.05) and between 44 and 50 hpf in STX-exposed embryos (*p* < 0.05 for *HSF2*, *p* < 0.01 for DUSP8). *NaV1.4b* gene expression increased significantly between 44 and 62 hpf in control embryos (*p* < 0.05). In exposed embryos, *NaV1.4b* gene expression was not detected during embryo development (between 26 and 44 hpf) but significantly increased between 50 and 62 hpf in lecithotrophic larvae (*p* < 0.05). The expression level of the *PPRC1* gene increased significantly between 50 and 62 hpf in control embryos (*p* < 0.05). In exposed embryos, *PPRC1* gene expression increased significantly between 44 and 50 hpf (*p* < 0.05) and then decreased significantly between 50 and 62 hpf (*p* < 0.05).

At 26 hpf, the expression of the *HSF2* and *NaV1.4b* genes was significantly lower in STX-exposed embryos compared to the control (*p* < 0.05), but no significant change was observed for the other genes (*p* > 0.05; Figure 2). At 44 hpf, only the expression of *NaV1.4b* was significantly different between the two treatments (*p* < 0.05). At 50 hpf, the expression levels of the four genes were significantly higher in STX-exposed larvae than in the control (*p* < 0.05). At 62 hpf, a significant decrease in *PPRC1* gene expression was observed in intoxicated larvae compared to that of the control (*p* < 0.05). In contrast, no significant changes were noted for the other genes (*p* > 0.05).

### 2.3. Enzyme Analyses

The specific activities of protein phosphatase (PP), alkaline phosphatase (ALP), and peroxidase (PER) were detected from 8 to 62 hpf in the control and from 26 to 62 hpf in the STX treatment (Figure 3). A significant decrease in the activity of both types of phosphatases was observed between 44 and 50 hpf (i.e., just after larval hatching) in both treatments (*p* < 0.01). PER activity was significantly higher at 26 hpf than at 50 hpf in the control (*p* < 0.05) and in the STX-exposed embryos (*p* < 0.01). Nevertheless, the activity of this enzyme did not change significantly (*p* > 0.05) between 26 and 44 hpf or between 44, 50, and 62 hpf (Figure 3C).

At 26 hpf, PP activity was significantly lower in STX-exposed embryos compared to control embryos (*p* < 0.05)*;* however, at 44, 50, or 62 hpf (*p* > 0.05; Figure 3B), no significant difference in the specific activities of the ALP and PER enzymes was observed in STX-exposed embryos compared to those of the control (*p* > 0.05), regardless of the sampling time (Figure 3A,C).

## 3. Discussion

### 3.1. Reduction of Hatching Percentage due to STX Exposure

The high hatching percentage (71.16 ± 1.97%) observed in the control indicates that the majority of eggs were able to complete embryonic development and hatch into larvae. In comparison, Brenta et al. [36] obtained a higher hatching percentage (78.3 ± 2.9%) in the control, which suggests that the eggs in our experiment were of slightly lower quality.

A very low hatching percentage (4.92 ± 0.80%) was observed for embryos exposed to the STX concentration of 100 µg L^−1^ (Figure 1C), which suggests a very high embryo mortality rate (>90%). Indeed, large quantities of dead embryos were observed at 44 hpf in the tanks with STX. Once again, these results differ from those of Brenta et al. [34], who employed an equivalent dose (100 µg L^−1^ STX eq.) of gonyautoxin (GTX), an analog of STX, and observed a 50% hatching rate of *S. rivoliana* eggs. Together with neosaxitoxin (NeoSTX), STX exhibits the highest toxicity among paralytic shellfish toxins, while GTX is less toxic (Table 1). This difference in toxicity explains why STX was able to lower the egg-hatching rate to a greater extent than GTX.

Brenta [36] also highlighted that under equivalent concentrations (90 µg L^−1^ OA eq. and 100 µg L^−1^ STX eq.), hatching was more affected by OA, which is responsible for diarrhetic shellfish poisoning, than by STX, suggesting that *S. rivoliana* embryos are more sensitive to OA than to GTX. Thus, it can be hypothesized that in embryos, toxicity depends on the roles targeted substrates play in cellular mechanisms and their relative importance. Within a given family of toxins, which exhibit the same modes of action with specific substrates, the elevated toxicity of some members could be related to molecular structures that increase binding affinity to certain molecular receptors [5,9].

Paralytic shellfish toxins have a common molecular structure and variable R1-4 functional groups that influence their net charge and polarity, increasing the binding ability of PSTs to ion channel sites [9]. At the R4 position, the carbamate moiety common to STX and GTXs confers higher toxicity than the N-sulfocarbamoyl moiety present in C1 and C2 toxins or the decarbamate moiety characteristic of dc-GTXs. On the other hand, the lower toxicity of GTXs compared to that of STX is due to the presence of HOSO_3_ groups in positions R2 and R3, which decreases ion channel binding affinity and, consequently, the toxicity of the molecule (Table 1). PSTs can be classified according to their relative toxicity with the toxic equivalency factor (TEF): STX > GTX3 > GTX2 > C2 > C1 > dcGTX3 > dcGTX2. Given that STX has a higher toxicity than its analogs, it seems clear that the embryo hatching rate was more strongly impacted by STX exposure in this study than by GTX2-3, C1-2, or dcGTX2-3 exposure [36]. However, the hypothesis that high embryo sensitivity to STX results from the ability of this toxin to cross the chorion and directly reach the embryo within the egg must be confirmed in future studies. STX exposure increases the expression of various apoptosis-related genes in bivalves [40,41] and fish [42]. Thus, the results suggest that the exposure of embryos to STX, either by simple external contact or by incorporating the toxin into the egg, activates cellular mechanisms related to apoptosis, triggering embryo death. Nevertheless, other factors besides STX exposure, such as culture conditions or egg quality, may also influence the sensitivity of fish embryos to toxins and help explain the drastic embryo mortality observed in the present study (Appendix A).

### 3.2. Heat Shock Protein Induction

*HSF2* gene expression significantly increased between 50 and 62 hpf in control larvae and 50 hpf in STX-exposed larvae (Figure 2A). These results suggest that STX exposure resulted in the early induction of the *HSF2* gene during larval development in *S. rivoliana*. The *HSF2* gene is a member of the heat shock factor (HSF) family, which are activators of heat shock protein (HSP) transcription [43,44]. Heat shock proteins act as molecular chaperones to assist misfolded proteins in stressed cells and are involved in the responses to multiple environmental stressors, including heat shock [45] and exposure to contaminants [46] or toxins [47,48]. Thus, *HSF2* induction could indicate the probable accumulation of misfolded proteins within cells exposed to saxitoxin, as well as other processes involved in morphogenetic changes. In this study, *HSF2* induction occurred early during development in STX-exposed larvae at 50 and 62 hpf. These gene expression results provide evidence for STX-related stress response in *S. rivoliana* larvae and suggest high transcription of HSP coding genes. Saxitoxin exposure has been reported to induce an increase in HSPs in Medaka (*Oryzias latipes*) embryos, although during later embryonic-larval stages (i.e., 16 days post-fertilization) [49] Appendix A. Similarly, in juvenile gilthead seabream (*Sparus aurata*), notable induction of HSP70 proteins was observed with PST exposure and warming [50].

In addition, *HSF2* gene expression increased during embryo development in the control (Figure 2A). This result highlights the multiple roles of HSP and HSF proteins that are synthesized in response to environmental stress and during normal embryonic development. Thus, HSF proteins and HSP play vital roles in providing protection from multiple environmental stressors during the larval phase, one of the most vulnerable life stages of fish development [51]. *HSF2* has also been shown to be involved in the constitutive transcription of HSP genes [52], which would explain its presence in both the control and STX-exposed embryos. Although the differences were not statistically significant, lower *HSF2* expression at 26 hpf and higher expression of the same at 62 hpf can be visually noted in the exposed embryos compared to those in the control.

### 3.3. Regulation of Voltage-Gated Sodium Channels via Gene Expression

The primary mode of action of STX is to block voltage-gated sodium (NaV) channels by binding to site 1 of the α-subunit that forms the channel pore [11]. To highlight the impact of STX on NaV channels at the molecular level, we studied the expression of the *NaV1.4b* gene, which encodes the α-subunit of the NaV1.4 isoform. This isoform promotes electrogenesis in non-neuronal cells (e.g., skeletal muscle cells) and non-excitable cells (e.g., endothelial cells and red blood cells) [53]. The results revealed an absence of *NaV1.4b* gene expression between 26 and 44 hpf in STX-exposed embryos, followed by a 120-fold increase in expression between 50 and 62 hpf in STX-exposed larvae compared to that of the control (Figure 2B). These results suggest that the STX-associated stress response occurs in two phases during embryo and larval development: (i) STX completely inhibits the expression of the *NaV1.4b* gene at the embryo stage and (ii) an up-regulation of this gene occurs during the larval stage in response to this inhibition.

By regulating Na^+^ flow across the cell membrane to initiate and propagate action potentials, NaV channels are responsible for the electrical excitability of cells and play crucial roles in ion regulation, neuromuscular communication, and digestion [54,55]. In addition, NaV channels are involved in regulating the cellular K^+^ flux [56], which is required for the continuous functioning of the sodium-potassium pump (Na^+^/K^+^ATPase). Therefore, the inhibition of the *NaV1.4b* gene observed before and after hatching (26 and 44 hpf) may be a way of preserving the ATP pool, given that pre- and post-hatching phases are critical periods during fish development and are energetically costly [57]. In a broader context, the over-expression of the *NaV1.4b* gene could protect the cellular machinery of young larvae, as Na^+^ ions play key roles in many cellular mechanisms (e.g., neurotransmission, osmoregulation, and immunity) [58,59]. It would be useful to study the adenylate energy charge (AEC) by estimating the variation in energy content (in terms of ATP, ADP, and AMP concentrations) throughout the development of STX-exposed embryos to assess the impact of toxic stress on cellular metabolism, especially in these energy-related pathways.

To our knowledge, no study has identified the different NaV channel isoforms found in *Seriola* species or, more generally, in fish of the family Carangidae. In the case of the NaV1.4 isoform, which is primarily found in skeletal muscle cells, amino acid substitutions in the alpha subunit have been reported to confer resistance to toxins with positively charged guanidinium, including saxitoxin (STX), tetrodotoxin (TTX), and their derivatives. In particular, these substitutions alter the pore structure of the channels and electrostatic interactions between the toxin and pore, thus preventing the toxin from binding to the channel. This phenomenon has been described in several pufferfish species, the blue-ringed octopus (*Hapalochlaena* sp.), and soft-shell clam (*Mya areneria*) [60]. In the future, it would be interesting to identify the different isoforms of the NaV channels in *S. rivoliana*, the way they are related to ontogenetic development, and their distributions within tissues. The sensitivity and resistance of the different NaV channels to guanidinium toxins should also be assessed in this species.

### 3.4. No STX-Induced Oxidative Stress

Specific peroxidase activity was detected as early as 8 hpf and then significantly decreased between 26 and 50 hpf in both control and STX-exposed embryos (Figure 3C). Peroxidases are antioxidant enzymes capable of degrading hydrogen peroxide (H_2_O_2_), a reactive oxygen species (ROS) that oxidizes lipids, DNA, and proteins. These enzymes are integral to the antioxidant defense systems of aquatic organisms and protect cell membranes and various molecules within cells [61]. In the present study, peroxidase (PER) activation suggests the presence of H_2_O_2_ in both control and STX-exposed embryos. However, PER activity was not significantly higher in exposed embryos than in the control. In adult fish, PSTs are known to induce oxidative stress, which can be observed in the modulation of antioxidant enzymes such as catalase (CAT), glutathione peroxidase (GPX), and superoxide dismutase (SOD) [49,62,63] Appendix A. The results of PER activity do not suggest that STX induced oxidative stress in fish embryos and larvae. The overproduction of ROS is importantly related to cytotoxicity and cell apoptosis [64].

### 3.5. Activation of Lipid Metabolism and Digestion-Related Pathways

The regulation of lipid metabolism was assessed by measuring the expression of the *PPRC1* gene, which encodes the peroxisome proliferator-activated receptor gamma (PPARγ) coactivator relative protein 1. *PPRC1* gene expression was detected as early as 8 hpf in control embryos and significantly increased between 50 and 62 hpf in control larvae (Figure 2C). PPARγ is not only a key component in adipogenesis and lipid storage, biosynthesis, and metabolism but also helps regulate inflammatory responses, cell differentiation, and cell proliferation [65,66,67]. Lipid metabolism, and, more specifically, the β-oxidation of fatty acids, is an essential metabolic pathway for the consumption of yolk reserves during embryo development and the development of lecithotrophic larvae, which depend on lipid droplets because these are the only endogenous reserves available after yolk sac depletion [68].

The results obtained in this study indicate a metabolic activation at 62 hpf (i.e., 26 h old larvae) and provide evidence of the consumption of the remaining endogenous reserves prior to mouth opening and the first exogenous feeding. Similarly, Brenta et al. [36] found an increase in lipase activity in control larvae between 0 and 3 days post-hatching (i.e., before the onset of exogenous feeding), which may be a mechanism to initiate the digestion of exogenous food. Given that PPARγ is also involved in cell differentiation and proliferation, the induction of the *PPRC1* gene at 62 hpf may also indicate tissue development in young lecithotrophic larvae. Thus, the *PPRC1* gene expression results provide evidence that 26 h old lecithotrophic larvae are metabolically more active than embryos.

The activation of digestion-related pathways was investigated using the activity of ALP. In both the control and STX treatments, ALP activity was detected throughout embryo development (from 8 to 44 hpf) and then dropped to near-zero levels at 50 and 62 hpf. ALPs are intrinsic membrane proteins that are involved in digestion, membrane transport, innate immunity, and bone mineralization [69,70,71]. Thus, the increase in ALP activity in embryos may, in part, indicate the need to regulate the activity of enzymes involved in these various mechanisms, including digestion. Indeed, although the digestive system is not complete and functional at the embryonic stage, some digestive enzymes, such as ALP and lipase, have been detected in fish embryos and could potentially play roles in the absorption of lipids from maternal yolk [36,37]. In contrast, during the early larval stages of both treatments, ALP activity was almost zero at 50 and 62 hpf, indicating that either these enzymes exhibited little or no involvement in the lipid metabolism of lecithotrophic *S. rivoliana* larvae or that they were inactivated via other phosphatases or degraded via external stress.

STX exposure triggered an up-regulation of the *PPRC1* gene at 50 hpf followed by down-regulation at 62 hpf in STX-exposed larvae (Figure 2C). However, no significant changes in ALP activity were observed between the control and STX treatments. The present results of *PPRC1* gene expression suggest that the response to STX exposure occurs in two phases in lecithotrophic larvae. The first phase is characterized as active lipid metabolism that results in the rapid consumption of yolk reserves, which reflects a short-term stress response to meet energetic demands. During the second phase, the inhibition of lipid metabolism indicates either an inability of larvae to maintain a high metabolic rate when stress is prolonged or the total depletion of endogenous energy reserves at 62 hpf. Considering the results of the present study and those of Le Du et al. [37] and Brenta et al. [36], the pathways involved in lipid metabolism and digestion in *S. rivoliana* larvae are more affected by OA+DTX-1, followed by STX and GTX and their analogs.

As a consequence of the inhibition of fatty acid β-oxidation, stress resistance, and immune functions may be impaired in fish. However, to our knowledge, this hypothesis has only been proposed in adult fish [72] but not in lecithotrophic larvae. This may be because the innate immune response of stress resistance is not well developed at this larval stage and only becomes active at the larval mouth opening. Lecithotrophic larvae are more vulnerable than embryos to toxins because they do not have a protective envelope (chorion) and are consequently more vulnerable to toxin exposure. Moreover, the mechanisms for STX detoxification may not yet be developed in the early larval stages of *S. rivoliana* and only become activated later in development. In the larval stages of marine fish, adaptive mechanisms for detoxification or sequestration may have evolved as proposed for Pacific herring (*Clupea harengus pallasi*) [73].

### 3.6. Regulation of Protein Phosphatases

Protein phosphatases (PPs) are phosphatases that reverse the action of protein kinases by dephosphorylating amino acids in proteins, primarily serine (Ser), threonine (Thr), and tyrosine (Tyr) residues. For example, protein Ser/Thr phosphatases remove phosphate from Ser and Thr residues in proteins [74]. Protein phosphatases have been well studied in ecotoxicology, especially PP1 and PP2A, which are known to be inhibited by marine toxins such as OA and microcystin-LR (MC-LR) [75]. In the present study, PP activity was detected in control embryos from 8 hpf onwards (i.e., before hatching; Figure 3A and Figure 4), which was also observed in a previous study of *S. rivoliana* embryos [35]. Similarly, Shi et al. [76] found pre-hatching phosphatase activity in olive flounder (*Paralichthys olivaceus*) embryos. In the present study, PP activity was detected throughout embryonic development (from 8 to 44 hpf) and then drastically decreased to near-zero levels at 50 and 62 hpf during the early larval stages. This inhibition of PP activity in both control and STX-exposed larvae may result from the regulation of these enzymes via ontogenetic processes, which could be genetically programmed, or oxidative stress-related processes in cells. To date, the inhibition of PP by oxidative stress has never been described in marine organisms. On the other hand, in some cell lines, it has been demonstrated that H_2_O_2_ is able to inhibit PP activity rapidly [77,78]. Although PP activity was significantly lower in exposed embryos than those in the control at 26 hpf, it is unlikely that STX inhibits the activity of PPs, as this inhibitory capacity is linked to a particular molecular structure (i.e., an extended aliphatic chain with a circular component), which is common to OA and MC-LR [72] but not to STX (Appendix A).

Dual-specificity phosphatases (DUSPs) are protein phosphatases that can dephosphorylate many key signaling molecules, including mitogen-activated protein kinases (MAPKs), which are associated with stress responses, immune responses, cell proliferation, and differentiation [79,80]. Through their dephosphorylation activity, DUSP8 phosphatases trigger the inactivation of stress-activated MAPK isoforms, c-Jun amino-terminal kinases (JNKs 1–3), and p38 MAPKs. By regulating the activity of both classes of MAPKs, DUSP8 plays roles in innate and adaptive immunity [81,82]. Therefore, DUSP8 activity must be properly controlled. The present results show that STX exposure during embryonic development induced an early up-regulation of *DUSP8* in the first larval stages, which could indicate the need to regulate stress-activated MAPK activities or, alternatively, to release additional inorganic phosphates to produce more energy.

Nevertheless, an uncontrolled up-regulation of *DUSP8* can strongly impact stress-activated MAPK signaling pathways and thus alter many key cellular processes. In the present study, the expression level was 120-fold higher in STX-exposed larvae than in control larvae at 50 hpf (*p* < 0.01), which may indicate the uncontrolled expression of the *DUSP8* gene due to STX exposure. Consequently, the MAPK signaling pathways may be altered, leading to the inhibition of cell proliferation and differentiation. This is a likely hypothesis because some studies have reported that STX can retard growth or induce abnormal growth in fish embryos and larvae [20,49], reducing body weight and size in adult fish [63]. The results of *DUSP8* gene expression do not reflect those of PP enzymatic activity. This discrepancy may be because the enzymatic assay employed in this study does not specifically target the activity of DUSP8 but measures all PPs, including PP1 and PP2A, which are known to be more prominently represented in cells than DUSP8 [75].

### 3.7. Implications of Other Factors in Embryonic and Larval Development

Although in vivo experiments are easier to implement and more reproducible than in situ studies, they have the disadvantage of being less relevant from an ecological point of view, as laboratory conditions cannot closely mimic natural conditions. The toxicity of STX at a concentration of 100 µg L^−1^ induced very high mortality in embryos. This observation is noteworthy because equivalent concentrations of PSTs may be present in the ecosystem during HABs. If true, this is a major problem for larviculture and fish reared in marine culture cages, as marine toxins can cause the massive mortality of larvae in hatcheries. Water filtration systems can control PST-producing dinoflagellate and cyanobacterial cells, but their toxins, which are dissolved in seawater due to their polar nature [83], cannot be controlled. Paralytic shellfish toxins are also highly thermostable and difficult to degrade, which results in notable bioaccumulation in tissues [84]. Although PSTs can be easily excreted via renal processes in fish due to their hydrophilic nature, they cannot be excreted in the same manner during early life stages when renal systems are not yet fully developed, making eggs and larvae highly vulnerable to HAB toxins [51]. The spatiotemporal overlap between fish spawning and HABs is critical to the success of fisheries recruitment and fish rearing. Thus, the sublethal effects of marine toxins during embryonic and larval life stages must be considered to improve the performance of fisheries and aquaculture programs.

Some authors, including Reis Costa et al. [39] and Roggatz et al. [85], have described a global climate change scenario in which the concomitant decrease in pH and increase in ocean temperatures elevate the bioavailability and toxicity of marine toxins, such as STX and TTX, negatively affecting ecosystems and human and animal health. This scenario has important implications for ecotoxicology and the chemical signals mediating interactions among marine species, such as foraging, reproduction, predation, and defense, with unknown consequences for ecosystem stability and vital ecosystem services [39].

## 4. Materials and Methods

### 4.1. Acquisition, Extraction, and Quantification of Marine Toxins

An STX FDA Reference Standard was obtained from the US National Institute of Standards and Technology (NIST, RM 8642). The reference material (RM) was saxitoxin dihydrochloride (CAS No. 35554-08-6) in a solution containing hydrochloric acid at a concentration of 5 mmol L^−1^ in 20% ethanol in water (volume fraction). The reference value for saxitoxin dihydrochloride is 103 ± 2 μg g^−1^. The STX FDA standard was provided by the Marine Toxins and Amino Acids Laboratory of CIBNOR (*Centro de Investigaciones Biológicas del Noroeste S.C.*). Saxitoxin activity and toxicity were calculated via mouse bioassay (MBA) following the recommendations described in the official protocol of the Association of Official Analytical Chemists [86]. To prepare the toxin solution, the original STX standard was concentrated via evaporation and then diluted in sterile saline solution (0.9% NaCl), as described by Le Du et al. [37].

### 4.2. Assessment of Embryonic Development and Larval Viability

#### 4.2.1. Experimental Protocol

Fertilized eggs were obtained from the natural spawning of *Seriola rivoliana* broodstock maintained in captivity under optimum conditions in collaboration with Ocean Era (formally Kampachi Farms) in CIBNOR. Eggs were collected with a 300 µm mesh bag and volumetrically counted to add ~15,200 eggs (i.e., 19 mL) to each 1 L glass jar (replicate). We began with three replicates containing 330 mL of seawater with embryos exposed to standard saxitoxin (100 μg L^−1^) and three controls containing 500 mL seawater without toxins. This concentration was chosen based on previous work with *S. rivoliana* embryos and larvae exposed to PSTs (GTX 2–3, dc-GTX 2–3, and C1–C2), demonstrating that this concentration is sufficient to impact embryonic and larval development without resulting in death [36]. This concentration was also chosen for its ecological relevance, as similar concentrations naturally occur during HABs. The seawater (38‰ salinity) was previously filtered (0.45 µm). All glass replicates were placed in a water bath (24.0 ± 0.4 °C) and homogenized with strong airflow to oxygenate the eggs, which were constantly mixed from top to bottom (Figure 4).

For each treatment (STX and control), two replicates were used to monitor embryonic development. For this purpose, 500 µL of eggs per replicate were collected at 8, 26, and 44 hpf and fixed in 500 µL of RNAlater^®^ (Thermo Fisher Scientific, Carlsbad, CA, USA) solution for gene expression analysis. Another 500 µL of eggs per replicate were collected at the same time for enzymatic assays. The samples were stored in 2 mL Eppendorf tubes (Eppendorf^®^ Premium U410, New Brunswick, NJ, USA) at −80 °C for molecular or enzymatic assays.

To monitor larval viability, the eggs from three replicates per treatment were incubated for 24 h and then distributed in three 2 L plastic tanks containing previously filtered seawater with a 1 µm Gaff bag. At 12 and 24 h after incubation (i.e., 50 and 62 hpf), 500 µL of larvae per replicate were collected to monitor viability with a light microscope coupled to a digital camera.

#### 4.2.2. Hatching Percentage

Given that hatching occurs at 36 hpf in *S. rivoliana* [38], the hatching percentage was assessed at 44 hpf for the control and STX treatment once the majority of viable embryos had hatched and after all embryonic developmental monitoring samples had been collected. For this purpose, the remaining eggs and larvae in the different tanks were collected and preserved in Davidson’s solution. The numbers of eggs and larvae contained in 1 mL aliquots were determined and reported based on the total sample volume. The hatching percentage (HP) was calculated using Equation (1):(1)$$HP=\frac{NL-NE}{NL+NE}\times 100,$$
where NL and NE are the number of larvae and eggs, respectively.

### 4.3. Gene Expression

#### 4.3.1. Primer Design

Oligonucleotides were designed in Primer 3 Plus v. 3.2.6 (https://dev.primer3plus.com/index.html (accessed on 2 February 2022)) from the target sequences derived from the transcriptome of *S. rivoliana* larvae [87]. Then, the self-complementarity of the selected primers was checked with Oligo Calc (http://biotools.nubic.northwestern.edu/OligoCalc.html (accessed on 2 February 2022)). The primers are listed in Table 2.

#### 4.3.2. RNA Extraction and cDNA Synthesis

TRizol^®^ RNA Isolation Reagent (Thermo Fisher Scientific, Waltham, MA, USA) was used according to the recommendations of the manufacturer to extract total RNA. Around 100 mg (103 ± 4 mg) of embryos or larvae were manually homogenized in 1 mL of TRizol using pestles. The concentration and purity of RNA were measured using a NanoDrop 2000 spectrophotometer (Thermo Fisher Scientific). RNA quantity was then visualized via electrophoresis in a 2.0% agarose gel with 1X Tris-Sodium Acetate EDTA (TAE) buffer to confirm integrity. Finally, an Improm II kit (Promega, Madison, WI, USA) was used to synthesize cDNA following the procedure provided by the manufacturer. The reaction was conducted with 1 µg of total RNA in a thermal iCycler (Bio-Rad, Berkeley, CA, USA).

#### 4.3.3. Real-Time PCR

To quantify the target genes in *S. rivoliana* embryos and larvae, a standard curve was constructed to observe the dynamic range of primer detection, verify the amplification efficiency (100% efficiency corresponds to a slope of −3.32), and select the dilution at which the samples should be quantified (1:5 dilution for the *NaV1.4b* gene and 1:10 dilution for the other genes). Once the dilution was set, the expression level of the target genes was quantified using SsoFastTM EvaGreen^®^ Supermix (Bio-Rad, Hercules, CA, USA). In each well, 5 µL of sample was mixed with 10 µL of reaction mix (0.15 µL of forward and reverse primers, 0.6 µL of MgCl_2_, 7.5 µL of SsoFast™ EvaGreen^®^ Supermix (Bio-Rad, Hercules, CA, USA) and 1.6 µL of water). Each assay measurement was performed in triplicate. The 18S ribosomal RNA gene was chosen as a reference to normalize the quantification cycles (Cq), which were analyzed according to CFX-Manager software algorithms (Bio-Rad, Hercules, CA, USA).

### 4.4. Protein Determination and Enzymatic Assays

Around 100 mg (103 ± 9 mg) of embryos or larvae were homogenized with 500 µL sterile deionized water in 2 mL Eppendorf PCR tubes containing grinding beads. Homogenization was performed using a FastPrep-24TM homogenizer 5G (MP Biochemicals, Santa Ana, CA, USA). Total soluble protein content was measured in the homogenates via photometry, as described by Bradford [88], using Bio-Rad Protein Assay dye reagent (Bio-Rad 500-0205) and bovine serum albumin (BSA, Sigma A7906, Madrid, Spain) as the standard. Protein and alkaline phosphatase activities were measured using a fluorometric method based on the protocol of Gee [89]. Enzyme kinetics were followed at a fluorescence wavelength of 460 nm with an excitation wavelength of 335 nm for 30 min (60 reads, 30 s each) at 30 °C. Peroxidase activity was measured via end-point spectrophotometry at 450 nm. Protein determination and enzyme activity were assessed using a 96-well plate spectrophotometer (Varioskan™, Thermo Fisher Scientific). Samples were assayed in triplicate in 96-flat bottom microplates and corrected with a blank (i.e., a sample was replaced by deionized water in the reaction mix). The following reaction mixes were employed.

Protein phosphatase: 10 μL of sample diluted 1:3, 140 μL of reaction solution (50 mM Tris-HCl, 11 mM MgCl_2_, 5 mM dithiothreitol, and 200 µg mL^−1^ serum albumin, pH = 7.0), and 50 µL of substrate solution (6, 8-Difluoro-4-methylumbelliferyl phosphate [DiFMUP] 200 µM) purchased from Molecular Probes (Eugene, OR, USA).

Alkaline phosphatase: 10 μL of sample diluted 1:3, 140 μL of reaction solution (100 mM Glycine, 1 mM MgCl_2_, and 1 mM ZnCl_2_, pH = 10.4), and 50 µL of substrate solution (DiFMUP 200 µM).

Peroxidase: 20 µL of sample diluted 1:3, 100 µL of reaction solution (0.2 M dibasic sodium phosphate, 0.1 M citric acid, one TMB [3,3′,5,5′-Tetramethylbenzidine, Sigma T8665], and 2 µL hydrogen peroxide for 40 mL of solution), and 50 µL of stop solution (40 mM sulfuric acid) added as soon as the mixture turned blue in the microplate wells.

### 4.5. Data Treatment and Statistical Analysis

Relative normalized gene expression data were obtained from CFX Manager™ v. 3.1 (Bio-Rad, Berkeley, CA, USA). Alkaline and protein phosphatase activities were calculated based on the slope of the linear regression obtained from the Abs = f(t) curve. Total enzyme activity was related to total soluble protein content to obtain the specific enzyme activity. Peroxidase activity was calculated using the following formula:(2)$$\frac{{Abs}_{450}\times 200}{Proteins\;mg\;{mL}^{-1}}$$

The final values for enzyme activity and gene expression correspond to the average of the experimental replicates. Considering the small sample size of each treatment (*n* = 2 or 3), extreme values were only eliminated from analyses when necessary.

Data are presented as mean ± standard error (SE) of 2 or 3 replicates for each representative treatment. The normality and homoscedasticity of the data were first checked using the Shapiro–Wilk and Levene tests, respectively. A parametric Student’s *t*-test was performed to compare the means of the hatching percentages between the control and STX treatments. Most data sets of gene expression and enzyme activity results exhibited non-normal distributions or significant differences in variance homogeneity. Thus, a non-parametric Kruskal–Wallis test was used for the overall comparison of means between groups, followed by Dunn’s post hoc test for multiple mean comparisons. The significance level was set to α = 0.05 for all statistical tests. Statistical analyses and results were performed and prepared in R v. 4.1.2 (R Core Team 2021).

## 5. Conclusions

The present study improves our understanding of the toxicological effects of STX during the early life stages of *S. rivoliana*. An STX concentration of 100 µg L^−1^ induced very high mortality in embryos and affected the expression of genes coding for *NaV1.4b* channels, the main target of this toxin, as well as the expression and activity of enzymes involved in a wide variety of cellular mechanisms—including stress responses (*HSF2*), lipid metabolism (*PPRC1*), and digestion (ALP)—and signaling pathways associated with cell proliferation and differentiation (dual-specificity MAP kinase phosphatases). No significant differences in the specific activities of the ALP and PER enzymes were observed in STX-exposed embryos compared to those in the control. For further studies, a proteomic approach in conjunction with transcriptomic analysis should be followed to determine which cellular pathways are most affected by STX exposure and which responses are primed at the molecular and cellular levels during different embryonic and larval development stages.

## Figures and Tables

**Figure 1 marinedrugs-21-00597-f001:**
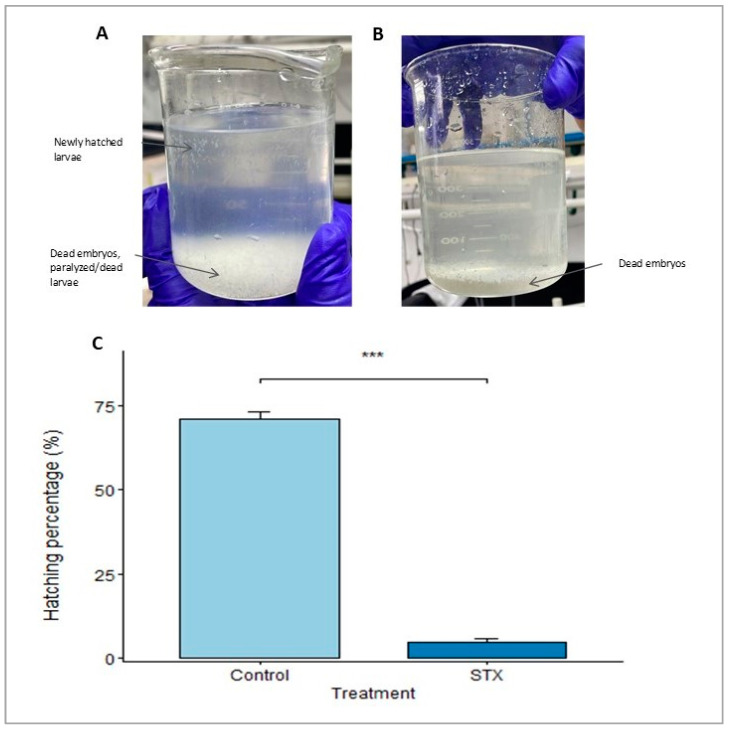
(**A**,**B**) Evaluation of hatching rate and larval viability at 44 hpf in the STX treatment. (**C**) Hatching percentages of *Seriola rivoliana* embryos. Control: embryos cultivated without toxins; STX: embryos exposed to saxitoxin standard solution (100 μg L^−1^ STX eq.). Asterisks (***) indicate significantly different means between the control and STX treatment (Student’s *t*-test, *p* < 0.001). Bars represent the mean ± standard error (*n* = 3).

**Figure 2 marinedrugs-21-00597-f002:**
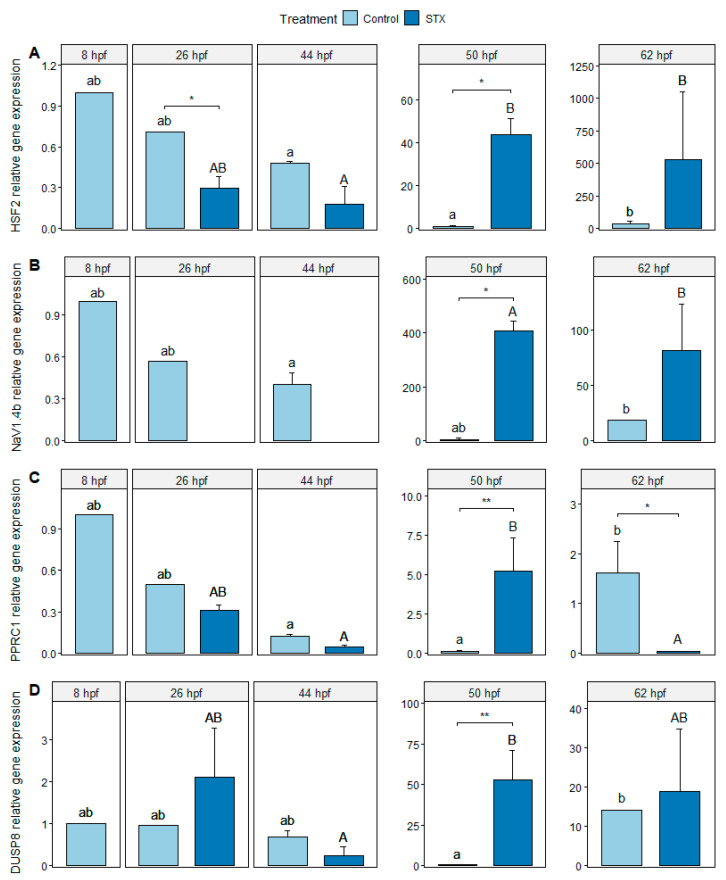
Relative gene expression (**A**) of heat shock factor 2 (*HSF2*), (**B**) NaV1.4 channel α-subunit (*NaV1.4b*), (**C**) *PPRC1* relative gene expression, and (**D**) dual specificity phosphatase 8 (DUSP8) genes from *Seriola rivoliana* embryos. Control: embryos cultivated without toxins; STX: embryos exposed to saxitoxin standard solution (100 μg L^−1^ STX eq.). Asterisks (* or **) indicate significant differences between treatments at 26, 50, and 62 hpf (Kruskal–Wallis *p* < 0.05 or *p* < 0.01, respectively). Letters ab and AB indicate significant differences between times under the control and STX (saxitoxin) treatments, respectively; (Kruskal–Wallis *p* < 0.05). Bars represent the mean ± standard error (*n* = 3).

**Figure 3 marinedrugs-21-00597-f003:**
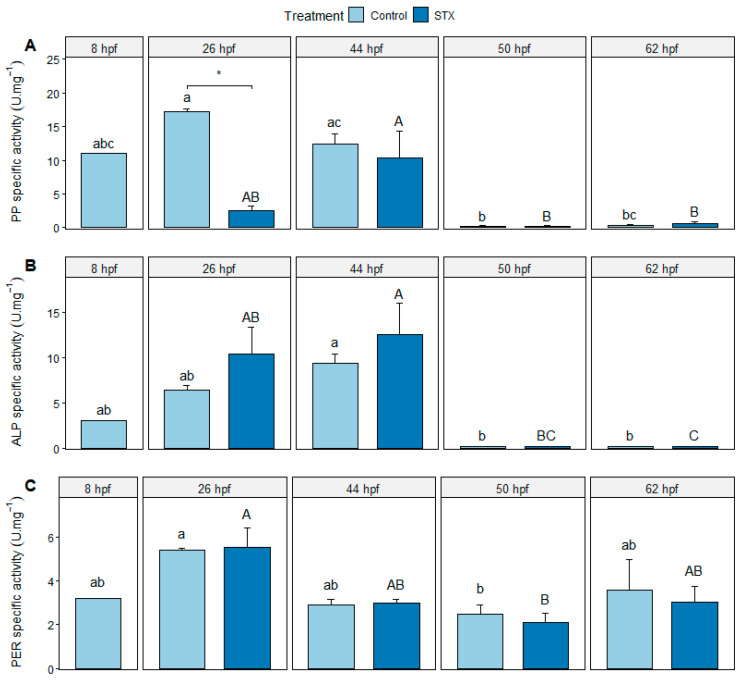
Specific activity (**A**) of protein phosphatase (PP), (**B**) alkaline phosphatase (ALP), and (**C**) peroxidase (PER) in *Seriola rivoliana* embryos. Control: embryos cultivated without toxins; STX: embryos exposed to standard saxitoxin solution (100 μg L^−1^ STX eq.). Results are expressed as units of enzyme per milligrams of total proteins (U mg^−1^). Asterisk (*) indicates significant differences between treatments at 26 hpf (Kruskal–Wallis *p* < 0.05). Letters ab, AB, c and C indicate significant differences in times under the control and STX (saxitoxin) treatment, respectively (Kruskal–Wallis *p* < 0.05). Bars represent the mean ± standard error (*n* = 3).

**Figure 4 marinedrugs-21-00597-f004:**
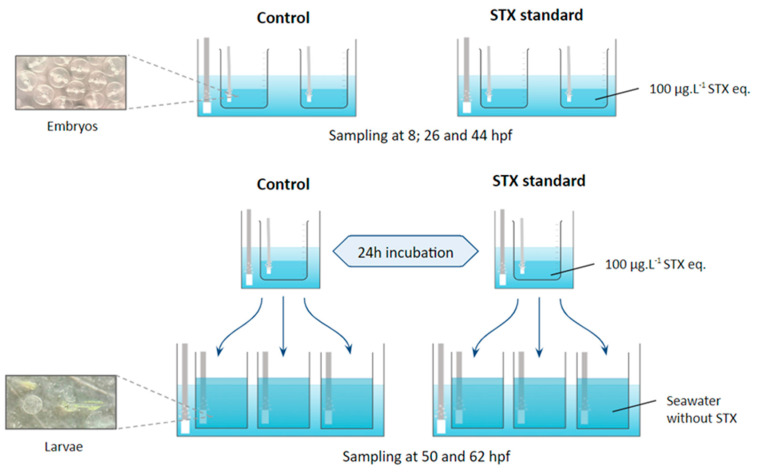
Experimental design for embryo culture and sampling.

**Table 1 marinedrugs-21-00597-t001:** Molecular structure and toxic equivalency factor (TEF) of the paralytic shellfish toxins (PSTs). Adapted from Leal and Cristiano [38] and Reis Costa et al. [39].

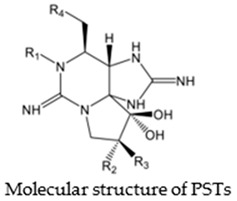					
**Toxin**	**TEFs ^1^**	**R_1_**	**R_2_**	**R_3_**	**R_4_**
** STX **	** 1.0 **	** –H **	** –H **	** –H **	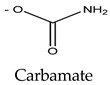
NeoSTX	0.92	–OH	–H	–H	
GTX1	0.99	–OH	–H	–HOSO_3_^−^	
** GTX2 **	** 0.36 **	** –H **	** –H **	** –HOSO_3_^−^ **	
** GTX3 **	** 0.64 **	** –H **	** –HOSO_3_^−^ **	** –H **	
GTX4	0.73	–OH	–HOSO_3_^−^	–H	
GTX5	0.064	–H	–H	–H	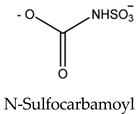
GTX6	–	–OH	–H	–H	
** C1 **	** 0.006 **	** –H **	** –H **	** –HOSO_3_^−^ **	
** C2 **	** 0.096 **	** –H **	** –HOSO_3_^−^ **	** –H **	
C3	0.013	–OH	–H	–HOSO_3_^−^	
C4	0.058	–OH	–HOSO_3_^−^	–H	
dcSTX	0.51	–H	–H	–H	–OH Decarbamate
dcNeoSTX	–	–OH	–H	–H	
dcGTX1	–	–OH	–H	–HOSO_3_^−^	
** dcGTX2 **	** 0.15 **	** –H **	** –H **	** –HOSO_3_^−^ **	
** dcGTX3 **	** 0.38 **	** –H **	** –HOSO_3_^−^ **	** –H **	
dcGTX4	–	–OH	–HOSO_3_^−^	–H	

^1^ Toxic equivalency factors (TEFs) were estimated from the specific activity in mouse unit (MU) per mmole determined by mouse bioassay (MBA). The analogs studied in *Seriola rivoliana* are indicated in blue.

**Table 2 marinedrugs-21-00597-t002:** Primer sequences used for quantitative real-time PCR analysis.

Gene Symbol	Target Gene	Primer	Sequence (5′–3′)
*18S*	18S ribosomal RNA	Forward Reverse	CTGAACTGGGGCCATGATTAAGAG GGTATCTGATCGTCGTCGAACCTC
*HSF2*	Heat shock factor protein 2	Forward Reverse	TTCATGGTGTTGGACGAGCA TGCTTGAAGTAGGGGTGCTG
*NaV1.4b*	Sodium channel protein type 4 subunit alpha B	Forward Reverse	TCCAGGACAACTCGAAAACC CGAAGTTGATCCAGTGCAGA
*PPRC1*	Peroxisome proliferator-activated receptor gamma coactivator-related protein 1	Forward Reverse	AACCCCAGCAAACACCTGAA ACACTTCCCATCTGCTGACG
*DUSP8*	Dual specificity phosphatase 8	Forward Reverse	CCTCACAGACAGGACACAACA GCTTTGGTGATGGTTTGACTG

## Data Availability

Data is contained within the article or Appendix A.

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
