# Peer review of "Toxicogenomic Effects of Dissolved Saxitoxin on the Early Life Stages of the Longfin Yellowtail (Seriola rivoliana)"

_marinedrugs, 2023, doi:10.3390/md21110597_

Round 1

Reviewer 1 Report

Comments and Suggestions for Authors

Some comment to the authors (not exhaustive)

Summary

Ligne 12 «Harmful algal blooms (HABs) result in the massive mortalities of marine animals due to toxin exposure and accumulation. »

Not all HABs cause animal deaths. For example, bivalves are able to accumulate STX from Alexandrium pacificum without dying.

Rewrite

Ligne 55 « The paralytic action of PSTs is due to their ability to block the influx of Na+ through voltage-gated sodium channels (VGSCs) by binding to site 1 of the α-subunit, thereby inhibiting the generation of action potentials in muscle and nerve cells [15,16]. In addition, STX has been shown to block calcium channels [17] and prolong gating in the potassium channels of heart muscle cells [18]. »

STX affects not only sodium channels but also other channels (calcium, potassium) as well as other proteins not involved in these channels.

Needs to be improved

Ligne 81 « The objective of the present study was to identify the effects of various PSTs, particularly STX »

You say you are going to study the effects of PSTs but you only looked exposure to  STX. no? Why did you choose to specifically test the effects of STX on S. riviolana?

Are these fish more exposed to STX than other PSTs in the environment?

Results

Ligne 112 Figure 1.

For easier reading I suggest you group the results of Figure S2 with Figure 1 

What does “hpf” mean? What are the larval mortality rates at 50 and 62 hpf?

Ligne 124. « The highest expression … »

We do not see this result in your figure 2. Please present the results of your controls differently.

Ligne 156 « similar to »

The expression of a gene is in no way similar to the activity of a protein. Even more these genes are different from those corresponding to the proteins studied. Reformulate

Ligne 158 « a significant drop»

What does it mean?

Here you present the results on the embryos and on the larvae as if the experiment were the same while for the larvae you carried out a second experiment by transferring your animals into water without STX. It is a little bit confused. To clarified, you must separate your results on embryos from those on larvae.

Same remark for the line 155

Discussion

Ligne 163 « Nevertheless, the activity of this enzyme did not change significantly (p > 0.05) between 26 and 44 hpf or between 44 and 50 hpf »

And the expression of PER at 62 hpf?

Ligne 200-208 «  PSTs can be classified according to their relative toxicity with  the Toxic Equivalency Factor (TEF): STX > GTX3 > GTX2 > C2 > C1 > dcGTX3 > dcGTX2. …  Given that STX has a higher toxicity than its analogues, it seems clear that the embryo hatching rate was more strongly impacted by STX exposure in this study than by GTX2-3, C1-2, or dcGTX2-3 exposure [34].»

The binding of STXs to ion channels and their toxicity are related to the sodium channel of human cells. Are the sodium channel structures of Seriola rivoliana cells similar?

« STX has a higher toxicity than its analogues »

What makes you say that? Do you have any information on the real toxicity of STX (compared to other derivatives) on Seriola rivoliana cells?

Ligne 213 « This hypothesis has been suggested by Le Du [35], who observed 60 to 100% mortality of S. rivoliana embryos exposed to the diarrhetic shellfish toxins OA and DTX-1 at concentrations of 120 and 175 μgL-1 OA eq., respectively »

Diabetic toxins are lipophilic toxins. Do you really think that the mechanism of action of hydrophilic toxins is the same as for lipophilic toxins?

Ligne 217 « Thus, the results suggest that the exposure of embryos to STX, either by simple external contact or by incorporating the toxin into the egg, activates cellular mechanisms related to apoptosis and triggers embryo death. »

Which of your results supports this hypothesis? You have not done Tunel cell marking and/or looked at the expression of pro- or anti-apototic genes.

Ligne 223 Table 1.

There is a problem of overlapping information in Table 1. Needs to be clarified.

The information in this table would be more appropriately presented in the introduction (state of the art).

Ligne 227 « HSF2 gene expression significantly increased between 50 and 62 hpf in control larvae and between 44 and 50 hpf in STX-exposed larvae »

I do not observe this in figure 2. HSF2 did not increased in 44 hpf in STX-exposed larvae? Needs to be corrected

Ligne 233 « Thus, HSF2 induction indicates an accumulation of misfolded proteins within cells »

There is nothing in your study to confirm this. Moreover, you suggest a multiple role of HSP  line 244 “This result highlights the multiple roles of HSP and HSF proteins”.

Ligne 254 « These results could have been significantly different between conditions if inter-replicate variability was lower »

I don't understand. Is there a problem with the design of your experiments?

Ligne 264 « These results suggest that the STX-associated stress response occurs in two phases during embryo and larval development »

How can you suggest that NaV expression at 50 and 62 hours is a response to STX stress since you transferred your larvae to toxin-free tanks?

Ligne 272 « Therefore, the inhibition of the NaV1.4b gene observed before and after hatching (26 and 44 hpf) may be a means of preserving the ATP pool, given that pre- and post-hatching phases are critical periods during fish development and are energetically costly [63]. »

If this were the case, why don’t we observe the same result for the control?

Ligne 312 « If STX-induced oxidative stress was not apparent with PER activity, this oxidative stress could also account for the high mortality observed at 44 hpf in STX-exposed embryos »

This is very speculative. Nothing in your results supports this hypothesis. Discard it

Ligne 354 « In comparison, Le Du [35] demonstrated the inhibitory effect of OA+DTX-1 on ALP while Brenta [34] observed a significant induction of lipase activity and gene expression by GTX and its analogues in 3-day-old larvae »

Here you compare lipophilic toxins, which have nothing in common with hydrophylic toxins apart from being toxic to humans. Off topic

Ligne 429 « For instance, the heat stress that occurred during the experiment could justify the induction of peroxidase activity in control embryos (Figure 3C). »

I did not find this very important information either in the math and method nor in the results. Please clarify this point? Needs to be clarrified

Math et Meth

Ligne 476. When did you put the STX in the tanks. At T0? Needs to be clarrified

Ligne 481. « STX eq. »

This is equivalent to toxicity for humans. As the effects are probably very different, it does not necessarily make sense for marine animals. Needs to be clarrified

 Ligne 481 Figure 1

Your first experiment was carried out on three control tanks and three STX tanks. There are only two tanks in Figure 4 at the top.  Needs to be clarrified

Ligne 498.

To monitor the larvae, you took samples at 50 and 62 hours. According to your protocol, you either did a 24-hour incubation with STX then took samples at 12 and 24 hours after transfer to another tank, therefore at 32 and 48 hours. Either you have carried out an initial incubation of 44 hours + a 24 hours extension with STX then taken the samples 12 and 24 hours after transfer, i.e. at 70 and 82 hours. In both cases dates does not match your given. Can you explain your experimental protocols better?

Ligne 501

What are the optical microscopy criteria that allowed you to define that an embryo or larva is alive or dead?

Ligne 531 Real time PCR on « target genes in S. riviolana ambryos » .

What about the larvae?

Reviewer 2 Report

Comments and Suggestions for Authors

In this manuscript, the authors report on a study on the toxicological effects of subacute exposure to saxitoxin (STX) during embryonic and early larval stages in Seriola rivoliana. Eggs were exposed to dissolved STX (100 μg STX eq L-1) and the toxic effect of STX was evaluated via the hatching percentage. The activity of three enzymes (protein and alkaline phosphatases and peroxidase) and the expression of four genes (HSF2, Nav1.4b, PPRC1, and DUSP8) was analyses. A very low hatching percentage (less than 5%) was observed in STX-treated embryos and low oxidative stress. On the other hand, the authors point out that STX induced regulation of the expression of the α-subunit of NaV channels (NaV1.4b), overexpression of all four candidate genes in STX-intoxicated lecithotrophic larvae. 

Major considerations:

1.- In the MS, there are sections that are too long, and should be summarized, especially the Discussion section.

2.- The paragraphs in the Introduction, lines 84-105, describe part of the results, and therefore, should be included in section 2 (Results).

3.- All signs (letters, symbols, etc.) in Figures 2 and 3 should be described in the figure captions.

4.- Protein phosphatases comprise an extensive family of proteins, please indicate which specific protein the authors refer to as PP.

5.- Discussion. 3.1. To compare toxic effects between STX and OA it must be done in molar concentrations. Due to the difference in the molecular weight, the comparison using ug L-1, it is a important mistake.

6.- Table 1 is not relevant in the discussion of results and could be included in the supplementary material.

7.- In general, the work is well executed although it has many limitations that are described in the conclusions section. The conclusions should be rewritten highlighting the achievements of this study.

8.- Table S3 should be included in the main text.

Minor considerations

9.- References 24, 37 and 38 are not cited.

10.- Table S4 is not cited. It does not contribute to the discussion and could be eliminated.

Reviewer 3 Report

Comments and Suggestions for Authors

Toxicogenomic Effects of Dissolved Saxitoxin on the Early Life Stages of the Longfin Yellowtail (Seriola rivoliana)

This manuscript explores the effect of dissolved saxitoxin, a PSP toxin, in early life stages of Seriola rivuliana, a well know Carangidae fish of commercial interest. The impact of PSP events in public health and marine fauna has been known for a long time, but advanced OMICS tools are allowing to explore deeper into the cellular mechanisms being affected. Here, the authors used advance transcriptomics technology to asses the impact of free saxitoxin in the transcription of 4 genes known to be particularly affected by saxitoxins.

Experiments were well designed. The only weakness, to have only 2 replicates per treatment in  some of the experiments. In these cases, please, justify that the statistical treatment used to test significance was appropriate.

The authors compared several times their results with those from a manuscript (reference # 34), still in submission phase and with common authors and research topics than the ms. presented here. Please, clarify what are the differences and the new findings in this paper for Toxins not contemplated in # 34.

ABSTRACT

Harmful algal blooms: This term is used in a vague and equivocal way. The authors should be more precise and mention “HABs producing toxins which are transferred through the food web (particulate toxin fraction) or are released in the water”.

INTRODUCTION

References 1 to 4 are cited without properly reading/understanding/reproducing the authors’ message.  In fact, ref #4 claims the increase of toxic event reports is largely due to increased monitoring and awareness. I made annotations in the WORD doc, but the authors should read more deeply and rewrite this paragraph.

The authors confuse the pathways (and effects) of particulate toxins (that require ingestion of bioaccumulated seafood) and free (“dissolved”) toxins in seawater. The effects are very different with particulate (e.g. direct human intoxication after eating contaminated shellfish, the main vector of PSP, DSP and ASP toxins) than with dissolved toxins. The dissolved fraction is detected in the marine environment with passive sampler (SPATT resins). In the case of early life stages, in particular before feeding, obviously the import toxin fraction is the dissolved fraction.

“For example, HABs have been responsible ……..released in the Gulf of California”

Here the authors only mentioned “released toxins” i.e., the so called “dissolved toxins”, but what kills adult marine mammals and birds is usually the ingestion of contaminated seafood and small pelagic fishes.

I was surprised/missed not to see any mention to:

·         Seriola rivoliana (70-73) , top carnivore in reef-associated food webs, as the main vector of Ciguatera fish poisoning in the Atlanctic (Macaronesian) Islands (Canarias and Madeira archipelagos) and the Azores.

·         Earlier studies on the effect of toxins on fish larvae, such as :

i.           Rountos et al 2019. Effects of the harmful algae, Alexandrium catenella and Dinophysis acuminata, on the survival, growth, and swimming activity of early life stages of forage fish. Mar. Environ. Res. 148, 46–56. https://doi.org/10.1016/j.marenvres.2019.04.013

ii.          Gaillard et al 2023. Mortality and histopathology in sheepshead minnow (Cyprinodon variegatus) larvae exposed to pectenotoxin-2 and Dinophysis acuminata. Aquatic Toxicology. 257. 106456. https://doi.org/10.1016/j.aquatox.2023.106456

·         Released toxins in seawater first detected with passive samplers (SPATT resins)

  RESULTS

Here and all through the manuscript, avoid mixing colloquial with scientific terms. In your experiment you have a control, different treatments (not “conditions”), and for each one you have several replicates (not “jars”).

Figure 2 legend 4th line “(n= 1 to 3)” If your n value is 1, how can you calculate means and standard errors?

DISCUSSION

Last para.: Avoid vague and too general statements if you are not able to explain in depth the argued mechanisms.

 Predictions for each species will vary under each specific climate-change scenario, combined with the characteristics of the local microalgal strains and the site-specific hydrodynamics.

This section is too wordy. Authors should be more concise, interpreting their results and being brief about other possible interpretations/hypothesis/speculations.

 In summary: A good work deserving publication (providing its uniqueness and differences from # 34) and after addressing this reviewer criticisms about vague statements on HABs and climate change just copied and pasted or even misinterpreted. 

Comments on the Quality of English Language

English is comprehensible but composition can be improved. Often the words order in the sentence follow a Spanish rather than English composition. Some sentences are too wordy. I made annotations in the doc. but needs further attention. 

Round 2

Reviewer 1 Report

Comments and Suggestions for Authors

minor revision.

See below 

Minor revision

 Line 19 «  Eggs were exposed to dissolved STX (100 μg STX eq L-1).

Replaced by

«  Eggs were exposed to dissolved 19 STX (100 μg L-1).

 Idem

Line 461 “ embryos exposed to standard saxitoxin (100 μg L-1 STX eq.) »

and line 462 « A concentration of 100 μg L-1 STX eq. was chosen based on previous …»

Replaced by

embryos exposed to standard saxitoxin (100 μg L’1) »

and by

«  This concentration was chosen based on previous…. »

Line 22 « Low hatching percentage (less than 5%) was observed in STX-treated embryos. »

Replace by

« A low hatching percentage (less than 5%) was observed in 44hpf embryos exposed to STX compare to 71% in unexposed control. »

Line 22 « The STX dose was too low to produce significant oxidative stress. « 

This sentence suggests that at a higher concentration, STX would induce oxidative stress.

I suggest this sentence

“At this STX concentration no oxidative stress in the ambryons was evidenced”

Line 24 « However, STX induced regulation of the expression of the NaV1.4 channel α-subunit (NaV1.4b)

Replaced by

«  However, STX induced the expression of the NaV1.4 channel α-subunit (NaV1.4b)

Reviewer 3 Report

Comments and Suggestions for Authors

The authors have addressed the points raised by this reviewer annd provided satisfactory answers.

My only objection is the use of the word epizootics to define some HAB effect. HABs are never equivalent to an infectious disease

Comments on the Quality of English Language

I have made only minor corrections in the annotated PDF, mainly related to:

i) sentences with the words in the "wrong" order, i.e. in an order that is not the conventional way in written English

ii) redundant words where the pronoun has been or could be used. For example, "embryos in the STX treatment and those embryos in the control treatment"  "those" is the pronoun substituting "embryos" (already mentioned in the sentence)

iii) After defining conditions used in the control, no need to write each "control treatment" . Enough to write "in the control"

iv) unnecesary repetition of words in the figure legends
